# Physical Therapy Assessment Tool Threshold Values to Identify Sarcopenia and Locomotive Syndrome in the Elderly

**DOI:** 10.3390/ijerph20126098

**Published:** 2023-06-10

**Authors:** Hae-In Kim, Myung-Chul Kim

**Affiliations:** Department of Physical Therapy, Eulji University, Seongnam 13135, Republic of Korea; khi920119@gmail.com

**Keywords:** sarcopenia, locomotive syndrome, physical therapy assessment tool, timed up and go test, berg balance scale, threshold value, elderly

## Abstract

This study aimed to evaluate sarcopenia and locomotive syndrome in Korean elderly patients, analyze the closely related factors, and determine the threshold for distinguishing participants with sarcopenia, locomotive syndrome, and non-disease. To this end, we enrolled 210 subjects aged 65 years or more and classified them into the sarcopenia (*n* = 36) and locomotive syndrome (*n* = 164) groups; a control group was also included (*n* = 10). We evaluated the characteristics of these patients using the Timed Up and Go (TUG) test and Berg Balance Scale (BBS) and performed statistical analysis. Our findings showed statistically significant differences between the groups, leading to the derivation of a significant threshold value. The threshold value of the TUG test between the control and locomotive syndrome groups was 9.47 s; the threshold value of the BBS was 54 points, respectively. The threshold value of the TUG test between the locomotive syndrome and sarcopenia groups was 10.27 s, and the threshold value of the BBS was 50 points, respectively. These findings suggest that sarcopenia is closely related to locomotive syndrome, and that sarcopenia and locomotive syndrome can be identified using a physical therapy diagnostic evaluation tool.

## 1. Introduction

One of the many effects of aging is physical weakness or a decline in physical function, which can directly impact the daily lives of patients. Aging of the brain reduces the number of receptors for neurotransmitters, including dopamine, acetylcholine, and serotonin, resulting in depression, memory impairments, and movement disorders [1,2]. As cognitive dysfunction chronically degenerates, memory declines, intelligence declines, and a slowing down of information processing occurs, leading to activity disorders [3]. Decreased proprioception, audiovisual and head position senses, and muscle strength lead to reflex and balance problems [4]. Therefore, various changes in the cognition and sensory organs due to aging limits the activities of daily living. This affects the quality of life by impairing the patient’s psychological and social functions [5].

In 2020, the Korea Institute for Health and Social Affairs reported that 12.2% of patients over 65 years of age reported functional limitations upon an assessment of activities of daily living; 4.3% of these respondents were aged 65 to 69 years, 8.1% were aged 70 to 74 years, and 41.9% were aged 85 years or more, respectively [6]. The aging process is often accompanied with an increase in functional limitations, which can significantly impact the lives of elderly patients. Therefore, it is critically important to accurately identify the condition of elderly patients and prevent functional decline. Sarcopenia and locomotive syndrome (LS) are two conditions of functional decline in elderly patients.

Sarcopenia is a major risk factor threatening the health and lives of elderly patients [7]. It refers to a decrease in skeletal muscle mass [8], including a decrease in muscle strength and a decrease in physical performance, both due to aging [9]. LS, a condition in which the musculoskeletal system deteriorates with aging, resulting in deteriorated motor functions and the requirement for nursing care, was first described by the Japanese Orthopedic Association (JOA). In this condition, the ability to move the body to stand or walk is reduced due to the aging of the musculoskeletal system [10].

Studies that jointly investigated and analyzed sarcopenia and LS are rare. However, according to a study by Kim et al., which investigated 237 Korean elderly people in 2022, it was found that all the elderly with sarcopenia among the study subjects had LS, and among the diagnostic evaluation items of the two diseases, a factor with a significant negative correlation, it was observed that the two diseases were closely related [11]. In addition, in a study by Nishimura et al., which surveyed 112 Japanese elderly people in 2020, the coexistence of sarcopenia and LS was associated with an increased risk of falls and a decreased ability to perform daily activities, suggesting similarities between the two diseases [12]. Sarcopenia and LS directly affect the decline in the physical functions of elderly patients, leading to various symptoms that threaten the health of these patients. Active prevention and management of these conditions are therefore necessary.

Sarcopenia and LS are diagnosed and evaluated using protocols proposed by the respective expert organizations. Initially, sarcopenia was diagnosed using skeletal muscle mass in the limbs divided by the square of the height (kg/m^2^) of less than 2.0 standard deviations of the mean of the younger population [13]. As such, sarcopenia is based on a decrease in muscle mass, and magnetic resonance imaging (MRI) [14], computed tomography enterography (CTE) [15], dual-energy X-ray absorptiometry (DXA), and bioelectrical impedance analysis (BIA) are used to measure the muscle mass in patients with sarcopenia [16]. Since then, several expert groups have suggested diagnostic criteria, including muscle strength and physical performance; in 2019, The Asian Working Group for Sarcopenia (AWGS 2019) presented an updated diagnostic evaluation criteria that included muscle mass, strength, and physical performance [16].

LS is assessed using the method described by the JOA. In 2012, the 25-question geriatric locomotive function scale (GLFS-25), a questionnaire used to evaluate LS, was presented [17]. Additionally, in 2015, the two-step test, stand-up test, and additional assessment protocols were introduced [18]. In 2020, the assessment protocol was supplemented and modified to assess the degree of LS at the first, second, and third stages [19].

As such, the diagnosis of sarcopenia and LS is based on the cut-off value of the assessment tool. In practice, in the clinical setting, these diagnostic assessments are often performed by physical therapists; thus, it is necessary to develop and validate assessment tools that can be used in the clinic by physical therapists to assess the body functions of patients.

Both sarcopenia and LS present a decrease in physical functions, often resulting from musculoskeletal issues. Given the complex nature of these conditions, it is essential to use physical therapy diagnostic assessment tools to identify the specific aspects of various bodily functions. However, few studies have used physical therapy diagnostic assessment methods for sarcopenia and LS. Assessment tools that can be used by physical therapists to provide an accurate clinical diagnostic evaluation are therefore necessary.

In this study, sarcopenia and LS were assessed in elderly Korean patients. Factors closely related to sarcopenia and LS were first identified by analyzing the physical characteristics, diseases, and physical functions of the patients, including functional mobility, gait speed, balance, and motor control. In addition, the general characteristics and threshold values for each factor of physical function were determined to distinguish between the participants with LS, sarcopenia, and no disease. The main aim of this study was to provide basic data that can be used to develop a physical therapy assessment method aimed at preventing and managing sarcopenia and LS in elderly patients.

## 2. Materials and Methods

### 2.1. Study Participants, Ethical Considerations, and Study Design

Participants aged 65 years, or more were recruited through the cooperation of the Seongnam Senior Industry Innovation Center located in Seongnam-si, Gyeonggi-do, Korea and the Hanam Misa Gangbyeon Comprehensive Welfare Center located in Hanam-si, Gyeonggi-do, Korea. A total of 240 participants were recruited, although 30 were subsequently excluded due to withdrawn consent, inability to walk independently, and inaccurate survey responses. Finally, the analysis included 210 participants. All participants were required to be able to communicate, read, and answer questionnaires independently. Participants were also required to be able to walk independently without the use of braces, assistive devices, or prosthetics. Participants with diseases that may have affected life indices (due to falls, dizziness, impaired blood pressure, or impaired respiration) during the functional evaluation, and those with visual or hearing impairments were excluded from the study.

This study was approved by the Eulji University Institutional Review Board (IRB) (approval number EU21-037). All participants provided informed consent for their participation in the study after the purpose and procedures were explained.

Our study was conducted as a cohort study to investigate the reality of sarcopenia and LS in Korea, but this study randomly selected subjects as a cross-sectional study and classified them into two experimental groups and a control group as a result of the screening test.

Each subject underwent a diagnostic evaluation for sarcopenia and LS followed by a physical therapy evaluation. Sarcopenia and LS were diagnosed according to the Sarcopenia Diagnostic Criteria presented by the 2019 AWGS and the 2020 Locomotive Syndrome Assessment Criteria presented by the JOA, respectively. Based on the diagnostic criteria proposed by the AWGS 2019, participants with a low muscle mass, low muscle strength, and/or low physical performance were diagnosed with sarcopenia. Additionally, the stand-up test, two-step test, and GLFS-25, which are assessment tools proposed by the JOA were used to diagnose LS. The Timed Up and Go test (TUG test) and the Berg Balance Scale (BBS) were used as physical therapy evaluation tools for the functional evaluation of the participants.

Participants diagnosed with sarcopenia were assigned to the sarcopenia group, while those who did not meet the criteria for sarcopenia but were assessed with LS were assigned to the LS group. Participants who were not diagnosed with either sarcopenia or LS were included in the control group. As a result, the sarcopenia group included 36 participants, the LS group included 164 participants, and the control group included 10 participants, respectively. All 36 participants in the sarcopenia group had LS. Table 1 presents the clinical characteristics of the participants.

### 2.2. Assessment Tools

#### 2.2.1. Sarcopenia Diagnostic Assessment

The Sarcopenia Diagnostic Criteria presented by the 2019 AWGS were used in this study [16]. The diagnostic factors for sarcopenia include a low muscle mass, low muscle strength, and low physical performance. Low muscle mass paired with a low muscle strength or low physical performance was diagnosed as sarcopenia. Low muscle mass paired with a low muscle strength and low physical performance was diagnosed as severe sarcopenia. In this study, the sarcopenia group included all patients with sarcopenia and severe sarcopenia.

Low muscle mass is defined as <7.0 kg/m^2^ for males and <5.7 kg/m^2^ for females, respectively. In this study, the combined skeletal muscle masses of the right arm, left arm, right leg, and left leg were measured using the InBody 570 Body Composition Analyzer (Inbody, Biospace, Republic of Korea). The total skeletal muscle mass of the limbs was divided by the square of the height to calculate the muscle index, which was used as the muscle mass in this study. Muscle strength was determined using grip strength measurements. Individuals with a muscle strength <28.0 kg in males and <18.0 kg in females, respectively, were considered to have a low muscle strength. The grip strength was measured twice in each hand, with the participant in an upright position using a spring-type digital grip dynamometer (my-5401, TAKEI, Japan), and the largest value of the two measurements was used in the subsequent statistical analyzes. Physical performance was assessed using the short physical performance battery (SPPB) in this study. A total score ≤9 points was defined as a low physical performance [16].

#### 2.2.2. LS Assessment

LS was assessed using the GLFS-25, stand-up test, and two-step test, as proposed by the JOA. LS was classified as stage 1 (LS 1: the stage at which the decline in mobility begins), stage 2 (LS 2: the decline in mobility is progressing and the risk of becoming unable to live independently is increasing), and stage 3 (LS 3: the stage at which the decline in mobility has progressed and interfered with the patient’s social life), respectively. If any of the three assessment tools met the criteria for each step, the participant was classified as having either LS 1, LS 2, or LS 3, respectively [19].

The GLFS-25 evaluates the pain and numbness, motor dysfunction, and immobility experienced by the elderly population during their daily activities. Drawing from the data of the previous month, this tool comprises a comprehensive set of 25 items, encompassing 4 items pertaining to pain, 16 items relating to daily activities and mobility, 3 items concerning social function, and 2 items addressing the topic of falls, respectively. It was translated into Korean and used through the consultation of Japanese physical therapy professors and experts who were fluent in Korean and Japanese; the questions for the translated version were found to be very reliable, with a Cronbach’s α confidence coefficient of 0.951. The stand-up test evaluates whether a person can stand up with both legs or one leg from a chair with heights of 40, 30, 20, and 10 cm, respectively. The two-step test measures the maximum stride length of the subject and evaluates it as the final value obtained by dividing it by the height. The evaluation equipment was self-made and used according to the standards suggested by the JOA.

The GLFS-25 was conducted as individual, in-person interviews, and the evaluator demonstrated the stand-up test and two-step test to each participant. The step-by-step decision criteria for LS are shown in Table 2.

#### 2.2.3. Body Function Assessment

##### TUG Test

The TUG test was used to measure the functional motility, mobility, and balance in this study. Boasting an impressive intra-rater reliability of 0.99 and an inter-rater reliability of 0.98, respectively, this physiotherapy function assessment tool serves as a valuable asset in clinical practice [20]. Functional movement damage is suspected when the TUG test time > 20 s. This tool has been used to assess the balance and functional movements in frail elderly patients and in patients with Parkinson’s disease [21] and is a predictor of sarcopenia in hospitalized elderly patients [22].

The TUG test was demonstrated by the evaluator, and the three-meter distance was marked using tape on the floor prior to the participant’s performance of the test. The TUG test begins with the participant sitting in a chair with armrests. Upon the start of the test, the participant rises from the chair and walks in a straight line for 3 m. The participant then turns around and returns to sitting in the chair. The time from the patient rising from the chair to the patient sitting back down was measured in this study [23].

##### BBS

In this study, the BBS was used to assess the fall risk and comprehensive balance abilities, such as posture maintenance and voluntary movement control [24]. The BBS consists of 14 items, including sitting, standing, and posture changes. Each item was scored from zero to four points, with higher scores indicating a more independent performance of the tasks. The maximum total score was 56. The inter-rater reliability and intra-rater reliability of the Korean version of the assessment tool used in this study were 0.97 [25].

### 2.3. Statistical Analysis

All statistical analyzes were conducted using SPSS statistical package (version 24.0, IBM, Albany, NY, USA). Statistical significance was set at *p* < 0.05. The chi-square test and Kruskal–Wallis test were performed to compare the characteristics of the three groups as appropriate based on the normality of the data. The threshold values of the characteristics and physical function assessment factors that were used to distinguish the control, LS, and sarcopenia groups were calculated using receiver operating characteristic (ROC) curves.

## 3. Results

### 3.1. Comparison of the Diagnostic Assessment Indicators and the Detailed Composition Ratio of LS between the Groups

#### 3.1.1. Comparison of the Sarcopenia Diagnostic Assessment Indicators

The means of the appendicular skeletal muscle mass (*p* < 0.05), skeletal muscle index (*p* < 0.05), grip strength (*p* < 0.05), and SPPB (*p* < 0.05) were found to be significantly different among the three groups. More specifically, the means of the appendicular skeletal muscle mass, skeletal muscle index, grip strength, and SPPB were found to be lowest in the sarcopenia group, followed by the LS and control groups, respectively (Table 3).

#### 3.1.2. Comparison of the LS Assessment Indicators

The mean of the GLFS-25 score was found to be significantly different among the three groups (*p* < 0.05). Furthermore, the results of the stand-up test (*p* < 0.05) and two-step test (*p* < 0.05) were found to be significantly different among the three groups. The mean score of GLFS-25 was highest in the sarcopenia group, followed by the LS and control groups, respectively. In the stand-up test and two-step test results, the LS 1 stage was the most common in the sarcopenia group, and LS 2 stage was the most common in the LS group; the control group was found to be without LS. (Table 4).

#### 3.1.3. Detailed Composition Ratio of Each Stage of LS

In the sarcopenia group, 41.67% patients had LS 1, 41.67% had LS 2, and 16.67% had LS 3, respectively. In the LS group, 66.46% patients had LS 1, 16.46% had LS 2, and 17.07% had LS 3, respectively. The composition ratio of each stage of LS was found to be significantly different among the groups (*p* < 0.05). (Table 5).

### 3.2. Body Function Assessment

The mean of TUG test time (*p* < 0.05) and BBS score (*p* < 0.05) were found to be significantly different among the groups. The mean of the TUG test was the highest in the sarcopenia group, followed by the LS and control groups, respectively. The mean of BBS total score was lowest in the sarcopenia group, followed by the LS group and control groups, respectively (Table 6).

### 3.3. Threshold Values

#### 3.3.1. Comparison of the Threshold Values of Body Function Assessment Factors between the Control and LS Groups

The threshold values of the body function assessment factors were able to distinguish between the patients with LS and the individuals with no disease (Table 7).

#### 3.3.2. Comparison of the Threshold Values of the Body Function Assessment Factors between the LS and Sarcopenia Groups

The threshold values of the body function assessment factors were able to distinguish between the patients with sarcopenia and LS (Table 8).

## 4. Discussion

This study assessed sarcopenia and LS in elderly Korean participants and established threshold values for each factor that can be used to classify patients as having no functional decline, LS, or sarcopenia. In the TUG test, threshold values of 9.47 s between the no functional decline group and the LS group, and 10.27 s between the LS and sarcopenia groups were calculated, respectively. In the BBS, threshold values of 54 s between the no functional decline group and the LS group, and 50 s between the LS and sarcopenia groups were calculated, respectively.

In this study, 24.36% males and 12.88% females were diagnosed with sarcopenia, respectively. Ko (2021), who measured the prevalence of sarcopenia using muscle mass and strength in elderly Koreans, reported that the prevalence of sarcopenia was 5.97% in males and 15.01% in females, respectively [26]. In addition, Kim and Won (2020), who measured the prevalence of sarcopenia using muscle mass, strength, and physical ability in Korean aged 70–84 years old, reported that the prevalence of sarcopenia was 21.3% in males and 13.8% in females, respectively [27].

Despite the use of the same diagnostic criteria, the prevalence of sarcopenia varied between the current study and previous studies. This may be due to the fact that participants with sarcopenia and severe sarcopenia were included in the sarcopenia group in this study. Since the criterium to evaluate the physical performance was to select one of three tests (SPPB, 6 m walk, and 5-time chair stand test, respectively), the choice of assessment tool may have affected the prevalence. Additionally, differences in the participant age between these studies may also have contributed to the differences in the reported sarcopenia prevalence.

All participants in this study, except for the control group, had LS. The prevalence of LS was found to be 95.24%. By gender, 91.03% of men and 97.73% of women had LS. The prevalence by stage was 59.05% for LS 1, 20.00% for LS 2, and 16.19% for LS 3, respectively. In a study by Taniguchi et al. (2021), which surveyed 2077 elderly people aged 60 years or older in Japan, the prevalence of LS 1 was 24.4%, LS 2 was 5.5%, and LS 3 was 6.5% [19], respectively, which was lower than the prevalence observed in the present study. This may be because we only assessed LS with the GLFS-25 questionnaire and included participants younger than 65 years of age.

In this study, all participants assigned to the sarcopenia group also had LS. As a result of the LS stage composition ratio survey by group, the prevalence of LS 1, which corresponds with mild LS, was 41.67% in the sarcopenia group and 66.46% in the LS group, respectively. The prevalence of LS 2 and LS 3, which corresponds to severe LS, was 58.34% in the sarcopenia group and 33.53% in the LS group, respectively. Therefore, participants in the sarcopenia group had more severe LS. This finding supports the hypothesis that when age-related functional decline occurs, LS may occur first, followed by sarcopenia. The results of a previous study reporting that LS can occur as a gradual deterioration to sarcopenia also supports this finding [11]. These observations also suggest that LS assessment tools can be used for the early diagnosis and prevention of sarcopenia.

In this study, the TUG test and BBS were used to assess the functional mobility and balance of the participants. The average time of the TUG test was 11.48, 10.34, and 8.14 s in the sarcopenia, LS, and control groups, respectively, with the longest observed in the sarcopenia group and the shortest in the control group. These results are consistent with those of a study by Kataoka et al. (2023) and Kim et al. (2020), who reported that participants with LS and sarcopenia had longer TUG completion times than participants without disease [28,29]. In addition, the average total score of BBS was 46.00, 51.01, and 54.90 points in the sarcopenia, LS, and control groups, respectively, i.e., the lowest in the sarcopenia group and the highest in the control group. These results are similar to those of Kim et al. (2022), who reported 48.47 points in the sarcopenia group and 51.14 points in the LS group, respectively [11]. These findings indicate that sarcopenia affects the coordination of body movements, resulting in balance disorders and the deterioration of gait function, and that LS is also closely related to dynamic balance.

To diagnose and assess sarcopenia and LS, threshold values have been suggested for each sex using various assessment tools; however, the threshold values for the body function assessment tool for elderly Korean patients remains unclear. In this study, the threshold values of the assessment tools that distinguished individuals with LS, sarcopenia, and no functional decline were determined.

In this study, the participants were divided into sarcopenia, LS, and control groups. The threshold values of the body function assessment tools were determined using ROC curves. Threshold values of the TUG test time and total BBS score were able to distinguish between the participants in each group. The TUG test time used to differentiate between the control group and the LS group was 9.47 s (AUC = 0.722), and that to differentiate between the LS group and the sarcopenia group was 10.27 s (AUC = 0.705), respectively. The total BBS score to differentiate between the control group and the LS group was 54 points (AUC = 0.736), and that to distinguish between the LS group and the sarcopenia group was 50 points (AUC = 0.699), respectively. A study of 68 hospitalized elderly people in Brazil reported that the cut-off point of the TUG test for predicting sarcopenia was 10.85 s [22], which is similar to the critical point of 10.27 s obtained in this study. In a study of 40 Koreans by Jung et al. (2020), they reported that the cut-off value of BBS for sarcopenia evaluation was 41 points or less [30]. Since this previous study targeted stroke patients, it would have been investigated with a value lower than the critical point of 50 points obtained in this study. The thresholds investigated in this study provide an improved understanding of the pathophysiological causes of sarcopenia and LS and can be used to research motor function decline and develop suitable interventions to prevent sarcopenia in elderly individuals.

This study has limitations in that it did not unify the number of samples between the groups and did not exclude gender differences between these groups. Future studies are recommended to supplement these points. In addition, since this study mainly targeted the elderly who were able to engage in social activities, a large-scale study is needed targeting both inpatients and the elderly who have difficulty in social activities.

This study is the first to identify the threshold of a common physical therapy assessment tool to differentiate sarcopenia from LS. The use of the proposed physical therapy diagnostic assessment tools will increase the efficiency of these diagnoses by reducing the time and physical limitations of the diagnostic assessments. In addition, the present results supplant previously used diagnostic assessment tools that have lost popularity and underscore the advantages of diagnoses carried out by physical therapists. Moving forward, the precision of diagnostic assessment and the therapeutic benefits of physical therapy are expected to enhance, allowing physiotherapists to make a greater contribution to patient healthcare.

## 5. Conclusions

The results of this study indicate that sarcopenia is closely related to LS, especially severe LS. The TUG test time was longest in the sarcopenia group and shortest in the control group, while the BBS scores were highest in the control group and lowest in the sarcopenia group, respectively. Additionally, threshold values for the short physical performance battery, TUG test time, and BBS score were determined to distinguish between individuals with sarcopenia, LS, and no disease.

The TUG test and BBS used in this study are physical therapy diagnostic evaluation tools commonly used by physical therapists to evaluate the functional abilities of the elderly in geriatric rehabilitation hospitals and nursing facilities for the elderly. In the future, it is suggested that these tools should be actively used in the evaluation of sarcopenia and LS as well as functional ability in the elderly. Furthermore these findings imply that physical therapy diagnostic assessment tools can be used to identify patients with sarcopenia and LS. The threshold values of the physiotherapy body function diagnostic assessment tools presented in this study will contribute to the development of physical therapy methods for the prevention and management of musculoskeletal disorders in elderly patients.

## Figures and Tables

**Table 1 ijerph-20-06098-t001:** General characteristics of the participants (*n* = 210).

Variables		Sarcopenia Group (*n* = 36)	LS Group (*n* = 164)	Control Group (*n* = 10)	*p*-Value
Gender ^1^	Male	19 (24.36%)	52 (66.67%)	7 (8.97%)	0.005 **
	Female	17 (12.88%)	112 (84.85%)	3 (2.27%)	
Age ^1^ (year)	60s	4 (7.02%)	49 (85.96%)	4 (7.02%)	0.145
	70s	28 (20.59%)	102 (75.00%)	6 (4.41%)	
	80s and over	4 (23.53%)	13 (76.47%)	0 (0.00%)	
Height (cm)		158.73 ± 8.97	159.03 ± 7.74	162.90 ± 5.19	0.172
Weight (kg)		57.06 ± 6.79	69.92 ± 8.87	61.62 ± 9.20	0.001 ** (a < b)

^1^ count (%). Values for all variables except ^1^ represent mean ± standard deviation. The chi-square test and Kruskal–Wallis test were significant at *p* < 0.01 **. LS, locomotive syndrome.

**Table 2 ijerph-20-06098-t002:** Determination criteria for each stage of locomotive syndrome.

Stage	Measurement	Action
LS 1	GLFS-25	A total score of 7 or more but less than 16
	Stand-up test	Can stand up with both feet from a 20 cm high chair, but unable to get up on one foot from a 40 cm high chair
	Two-step test	Values greater than or equal to 1.1 and less than 1.3
LS 2	GLFS-25	A total score of 16 or more but less than 24
	Stand-up test	Can stand up with both feet from a 30 cm high chair, but unable to get up on both feet from a 20 cm high chair
	Two-step test	Values greater than or equal to 0.9 and less than 1.1
LS 3	GLFS-25	A total score of 24 or more
	Stand-up test	Unable to get up with both feet from a 30 cm high chair
	Two-step test	Value less than 0.9

GLFS-25, 25-question geriatric locomotive function scale; LS, locomotive syndrome.

**Table 3 ijerph-20-06098-t003:** Comparison of the sarcopenia diagnostic assessment indicators among the groups.

Categories	Sarcopenia Group (*n* = 36)	LS Group (*n* = 164)	Control Group (*n* = 10)	*p*-Value
ASM (kg)	15.29 ± 3.09	17.24 ± 3.54	19.33 ± 3.08	0.001 **
SMI (kg/m^2^)	6.00 ± 0.66	6.75 ± 0.84	7.25 ± 0.83	0.000 ***
Grip strength (kg)	22.25 ± 7.25	24.68 ± 7.88	31.60 ± 5.92	0.001 **
SPPB (score)	8.78 ± 1.79	10.10 ± 1.70	11.50 ± 0.53	0.000 ***

Values represent mean ± standard deviation; The Kruskal–Wallis test was significant at *p* < 0.01 **, *p* < 0.001 ***. ASM, appendicular skeletal muscle mass; LS, locomotive syndrome; SMI, skeletal muscle index; and SPPB, short physical performance battery.

**Table 4 ijerph-20-06098-t004:** Comparison of the LS assessment indicators among the groups.

Categories		Sarcopenia Group (*n* = 36)	LS Group (*n* = 164)	Control Group (*n* = 10)	*p*-Value
GLFS-25 (score) ^1^		12.69 ± 14.47	10.00 ± 12.42	2.20 ± 1.87	0.009 **
Stand-up test group ^2^	None	1 (2.78%)	12 (7.32%)	10 (100.00%)	0.000 ***
	LS 1	27 (25.00%)	133 (81.10%)	0 (0.00%)	
	LS 2	5 (13.89%)	13 (7.93%)	0 (0.00%)	
	LS 3	3 (8.33%)	6 (3.66)	0 (0.00%)	
Two-step test group ^2^	None	8 (22.22%)	52 (31.71%)	10 (100.00%)	0.000 ***
	LS 1	16 (44.44%)	79 (48.17%)	0 (0.00%)	
	LS 2	10 (27.78%)	22 (13.41%)	0 (0.00%)	
	LS 3	2 (5.56%)	11 (6.71%)	0 (0.00%)	

Values represent mean ± standard deviation; The chi-square test and Kruskal–Wallis test were significant at *p* < 0.01 **, *p* < 0.001 ***. GLFS-25, 25-question geriatric locomotive function scale; and LS, locomotive syndrome. ^1^ is the average score for the group and ^2^ is the number (proportion) of eligible subjects in the group.

**Table 5 ijerph-20-06098-t005:** Detailed composition ratio of each stage of LS among the groups.

Categories	Sarcopenia Group (*n* = 36)	LS Group (*n* = 164)	*p*
LS 1	15 (41.67%)	109 (66.46%)	0.003 **
LS 2	15 (41.67%)	27 (16.46%)	
LS 3	6 (16.67%)	28 (17.07%)	

Values are count (%); The chi-square test was significant at *p* < 0.01 **. LS, locomotive syndrome.

**Table 6 ijerph-20-06098-t006:** Comparison of body function assessment among the groups.

Categories	Sarcopenia Group (*n* = 36)	LS Group (*n* = 164)	Control Group (*n* = 10)	*p*-Value
TUG test (sec)	11.48 ± 2.41	10.34 ± 5.60	8.17 ± 1.07	0.000 ***
BBS (score)	46.00 ± 10.54	51.01 ± 6.30	54.90 ± 1.20	0.000 ***

Values represent mean ± standard deviation; The Kruskal–Wallis test was significant at *p* < 0.001 ***. BBS, Berg Balance Scale; LS, locomotive syndrome; and TUG, Timed Up and Go test.

**Table 7 ijerph-20-06098-t007:** Comparison of the threshold values of body function assessment factors between the control and LS groups.

Categories	Cut-Off	AUC	SEN	SP	*p*
TUG test (sec)	9.47	0.722	51.29	74.47	0.004 **
BBS (score)	54	0.736	90.87	24.10	0.012 *

Values represent mean ± standard deviation; The ROC (receiver operation characteristic) curve was significant at *p* < 0.05 *, *p* < 0.01 **. AUC, area under curve; BBS, Berg Balance Scale; SEN, sensitivity; SP, specificity; and TUG, Timed Up and Go.

**Table 8 ijerph-20-06098-t008:** Comparison of the threshold values of the body function assessment factors between the LS and sarcopenia groups.

Categories	Cut-Off	AUC	SEN	SP	*p*-Value
TUG test (sec)	10.27	0.705	65.74	54.00	0.000 ***
BBS (score)	50	0.699	79.73	35.42	0.000 ***

Values represent mean ± standard deviation; The ROC (Receiver Operation Characteristic) curve was significant at *p* < 0.001 ***. AUC, Area Under Curve; BBS, Berg Balance Scale; SEN, Sensitivity; SP, specificity; TUG, Timed Up and Go.

## Data Availability

The data presented in this study are available upon request from the corresponding author.

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
