# Peer review of "Physical Therapy Assessment Tool Threshold Values to Identify Sarcopenia and Locomotive Syndrome in the Elderly"

_ijerph, 2023, doi:10.3390/ijerph20126098_

Round 1
Reviewer 1 Report
The paper presents valuable insights into assessing and classifying sarcopenia and locomotive syndrome (LS) in elderly Korean participants. The authors have made a commendable effort to address an important topic and have conducted a well-designed study to investigate the relationship between these conditions.
The paper comprehensively analyzes sarcopenia and LS, presenting novel threshold values for the TUG test and BBS scores to classify patients with no functional decline, LS, or sarcopenia. The methodology employed in the study appears sound, and the results demonstrate clear distinctions between the groups. The discussion and conclusion sections effectively interpret the findings, highlighting the implications for clinical practice and the potential role of physical therapy diagnostic assessment tools in early diagnosis and prevention.
While the paper contains several strengths, I would like to provide some constructive feedback and suggestions to enhance its quality and clarity:
1. In the introduction, it would be beneficial to provide a more detailed background on sarcopenia and LS, including their definitions, epidemiology, and associated consequences in the aging population. This will help readers better understand the significance of the study and its contribution to the field.
2. The methods section could benefit from additional information on the participant selection process, including any exclusion criteria and the rationale behind the sample size calculation. This will improve the transparency and reproducibility of the study.
3. Providing more specific references to the previous recent studies cited in the discussion section is important. Including authors' names, publication years, and key findings will help readers locate and understand the relevant literature more easily.
4. In the conclusion section, it would be valuable to elaborate on the practical implications of physical therapy diagnostic assessment tools and how physiotherapists can implement them in patient care. This will give readers a clearer understanding of the study findings' potential benefits and practical applications.
Based on the strengths and improvements suggested, I think the manuscript should undergo minor revisions. The authors have demonstrated a solid understanding of the subject matter and have presented their findings coherently and logically. Addressing the suggestions above will enhance the manuscript's clarity and impact.
Thank you for considering my evaluation of this manuscript. I appreciate the opportunity to review it, and I am confident that this paper will contribute to the journal with the suggested revisions. I am available for any further clarification or discussion regarding this review.
Author Response
Dear Reviewer
Thank you for your detailed review comments on our paper.
We sincerely responded to the opinions of the reviewers.
Please check the file below for your response to the review opinion.
Thank you.

Reviewer 2 Report
The MS “Physical therapy assessment tool threshold values to identify sarcopenia and locomotive syndrom in the elderly” deals with the important task of accurate identification of the actual physical condition of elderly patients. The study demonstate that sarcopenia is closely related to locomotive syndome (LS). Several criteria are suggested, such as TUG, Timed Up and Go, test and BBS, Berg Balance Scale, and threshold values are found so that the physical therapy methods could be used and, with this knowledge, further developed for the prevention and/or treatment of muscle disfunction in the elderly.
The study is well planned and performed. My only concern is the number of patients in the Control group (n=10) , while the groups with disfunctions appear significantly larger: ‘Sarcopenia’ (n = 36) and LS (n = 164). The reason for such a small group should be indicated and the limitations discussed.
A diagram showing wide enough distribution of some most important to the authors view parameters in the control group can be helpful.
Otherwise the use of statististical methods presuming Gaussian distribution of the parameter (that usually implies quite large number of points) are under question.
Another point: in some studies of the physical characteristrics of the elderly the difference in TUG test time was recorded between male and female patients (e.g. Ibrahim et al., 2017, https://www.ncbi.nlm.nih.gov/pmc/articles/PMC5626462/). Was it the case in this study? How variable were the characteristics?
Minor points:
Abbreviations AUC, SEN, and SEP in the Tables 7 and 8 are not spelled out.
Author Response

(The authors gave the same response as above.)

Reviewer 3 Report
Dear Authors,
In this study you aimed at evaluating sarcopenia and locomotive syndrome in Korean elderly, analyzing the closely related factors, and determining the threshold for distinguishing participants with sarcopenia, locomotive syndrome, and non-disease.
As you described in this paper, sarcopenia and locomotive syndrome present a decrease in physical functions, often resulting from musculoskeletal issues, but assessment tools that can be used by physical therapists to provide an accurate clinical diagnostic evaluation are still necessary.
You find that sarcopenia is closely related to locomotive syndrome and that sarcopenia and locomotive syndrome can be identified using a physical therapy diagnostic evaluation tool.
I appreciate your work, and the topic is very interesting.
However, I suggest some minor revision to improve your paper.
INTRODUCTION
Please describe the different tools to analyze muscle wasting, including DEXA, BIA, CT an MRI.
Reference suggested: “Nardone OM. Impact of Sarcopenia on Clinical Outcomes in a Cohort of Caucasian Active Crohn's Disease Patients Undergoing Multidetector CT-Enterography. Nutrients. 2022 Aug 23;14(17):3460. doi: 10.3390/nu14173460”.
Author Response

(The authors gave the same response as above.)
